# Improving MRAM Performance with Sparse Modulation and Hamming Error Correction

**DOI:** 10.3390/s25134050

**Published:** 2025-06-29

**Authors:** Nam Le, Thien An Nguyen, Jong-Ho Lee, Jaejin Lee

**Affiliations:** 1Department of Information Communication Convergence Technology, Soongsil University, Seoul 06978, Republic of Korea; namltb@soongsil.ac.kr (N.L.); anthienng1995@soongsil.ac.kr (T.A.N.); 2School of Electronic Engineering, Soongsil University, Seoul 06978, Republic of Korea; jongho.lee@ssu.ac.kr

**Keywords:** asymmetric write error rate, cascaded channel, error correction codes (ECCs), the Hamming code, non-volatile RAM, sparse codes, spin-torque transfer magnetic random-access memory (STT-MRAM)

## Abstract

With the rise of the Internet of Things (IoT), smart sensors are increasingly being deployed as compact edge processing units, necessitating continuously writable memory with low power consumption and fast access times. Magnetic random-access memory (MRAM) has emerged as a promising non-volatile alternative to conventional DRAM and SDRAM, offering advantages such as faster access speeds, reduced power consumption, and enhanced endurance. However, MRAM is subject to challenges including process variations and thermal fluctuations, which can induce random bit errors and result in imbalanced probabilities of 0 and 1 bits. To address these issues, we propose a novel sparse coding scheme characterized by a minimum Hamming distance of three. During the encoding process, three check bits are appended to the user data and processed using a generator matrix. If the resulting codeword fails to satisfy the sparsity constraint, it is inverted to comply with the coding requirement. This method is based on the error characteristics inherent in MRAM to facilitate effective error correction. Furthermore, we introduce a dynamic threshold detection technique that updates bit probability estimates in real time during data transmission. Simulation results demonstrate substantial improvements in both error resilience and decoding accuracy, particularly as MRAM density increases.

## 1. Introduction

Memory technology plays a pivotal role in modern computing systems. With the increasing proliferation of smart sensors and edge computing, there is a growing demand for memory solutions that support frequent write operations, low power consumption, and fast access times [1,2]. Conventional memory technologies, such as dynamic random-access memory (DRAM) and synchronous DRAM (SDRAM), are constrained by their volatility and relatively high energy requirements. In contrast, magnetic random-access memory (MRAM) has emerged as a promising non-volatile alternative, offering high-speed access, reduced power consumption, and superior endurance key attributes for sensor development [3,4].

Owing to these advantages, MRAM has been considered a candidate for unified embedded non-volatile memory (NVM) in Internet of Things (IoT) devices [1]. Notably, its energy efficiency directly addresses the limited battery life of sensors in wireless sensor networks. Among various MRAM technologies, spin-transfer torque MRAM (STT-MRAM) is particularly noteworthy, as it combines the speed and durability of DRAM with the non-volatility of flash memory, making it highly suitable for next-generation computing applications [1,5,6].

In STT-MRAM, digital data are managed via an nMOS transistor, which regulates current flow. The data themselves are stored as resistance values within a magnetic tunnel junction (MTJ). The MTJ comprises two ferromagnetic layers separated by a thin tunneling oxide barrier. These layers include a fixed “reference layer” and a “free layer”, the magnetization of which can be altered by a spin-polarized current. When the magnetization of the free layer is antiparallel to that of the reference layer, the MTJ exhibits high resistance. Conversely, when the magnetizations are aligned in parallel, the resistance is low [7].

STT-MRAM offers several compelling advantages, positioning it as a leading contender in the non-volatile memory (NVM) market [8]. However, similar to other established memory technologies, it faces critical technical challenges that can significantly affect its performance and reliability [9]. One of the primary concerns is process variation, arising from inconsistencies in the geometry and resistance of the magnetic tunnel junction (MTJ) storage elements [10]. These variations disrupt access operations, resulting in deviations in MTJ conductance and fluctuations in the switching threshold current.

Synchronization issues further impact STT-MRAM reliability. In synchronous designs, write operations are typically confined to fixed time windows, which may prevent certain memory cells from completing their state transitions within the allotted duration [8]. Additionally, thermal fluctuations introduce further instability by injecting randomness into the switching behavior. These circuits use MTJ-based sensors and hybrid designs that include self-referencing mechanisms or poly-resistor references to stabilize read operations under changing temperature conditions [11]. This increases the likelihood of unintended resistance changes in the MTJ, even in the absence of explicit write operations [12,13]. Another critical issue is read disturbance, wherein the read operation itself can inadvertently cause MTJ switching, leading to both read and write errors and thereby compromising data integrity. These reliability concerns are further exacerbated by manufacturing imperfections, which can degrade overall device performance. As technology nodes scale down, these challenges are expected to worsen, necessitating advanced design methodologies and process-level optimizations to improve STT-MRAM robustness and efficiency.

To mitigate these challenges, a variety of circuit-level and architectural solutions have been proposed. For example, the study in [14] recommended using transistors smaller than the worst-case size to reduce write energy consumption. It further advocated a hybrid magnetic and circuit-level design approach to enhance STT-MRAM efficiency. Likewise, the authors in [15] proposed a hybrid memory architecture that integrates static RAM (SRAM) with STT-MRAM, aiming to balance performance with energy efficiency. Another study [16] investigated the use of write buffers to improve overall system throughput.

In the domain of signal processing, advanced channel coding plays a critical role in improving system performance and addressing the aforementioned reliability concerns. The increasing demand for high-density storage and cost-effective digital integrated circuits (ICs) has accelerated the adoption of sophisticated coding and signal processing techniques in modern memory systems [8]. Errors occurring during read and write operations can be mitigated through the application of error-correcting codes (ECCs). For instance, the use of (71, 64) Hamming codes and (72, 64) extended Hamming codes has been proposed for STT-MRAM to correct single-bit errors efficiently [17]. Furthermore, BCH codes with multi-bit error correction capabilities have been introduced in [18] to enhance storage density and robustness against more severe fault conditions.

Channel modeling has been extensively utilized to develop and evaluate error-correcting codes (ECCs) for STT-MRAM systems [17,19,20,21,22]. Among these models, the cascaded channel model proposed in [17] has proven effective in facilitating error-rate simulations and in serving as a representative communication channel model. Building upon this model, Nguyen [8] observed that errors in STT-MRAM occur independently, with an asymmetric probability distribution between logic 0 and logic 1 bits.

To address this asymmetry, the authors in [8] introduced a 7/9 sparse coding scheme designed to mitigate the imbalance in bit error probabilities. This method constrains the Hamming weight of each output codeword to remain below half of the total codeword length, thereby reducing the likelihood of erroneous bit flips. However, when attempting to improve the code rate by increasing the codeword length, encoding sparse codes typically relies on mapping strategies based on Euclidean distance decoding, an approach that is computationally intensive and requires substantial memory resources.

To overcome these limitations, the authors in [23] proposed an alternative encoding method that employs a generator matrix to directly produce sparse codewords with a minimum Hamming distance (MHD) of three. Additionally, they developed a corresponding decoder based on syndrome decoding algorithms, which significantly reduced complexity and improved overall efficiency compared to the mapping-based techniques used in earlier studies. Nevertheless, the effectiveness of their syndrome decoder was limited by its assumption of an equal distribution of 0s and 1s in codewords during threshold detection. This assumption proved problematic in sparse coding contexts, ultimately restricting the decoder’s accuracy and performance.

To overcome the aforementioned limitation, we enhance the encoder and propose a novel method to improve the syndrome decoder by refining threshold detection through the dynamic and automatic estimation of bit-0 and bit-1 probabilities within the codeword. Specifically, during the encoding process, three additional bits—referred to as sparse check bits—are appended to the user data. The extended data are then processed using a generator matrix to produce the final codeword. If the generated codeword does not satisfy the sparsity constraint, it is inverted to ensure compliance with the sparse code characteristics.

On the decoding side, the received signal is processed using a syndrome decoding algorithm. The sparse check bits are utilized to detect whether the codeword was inverted during encoding, thereby enabling accurate data reconstruction. Furthermore, to support adaptive and accurate threshold detection, the initial probabilities of bit-0 and bit-1 in the codeword are computed prior to transmission. These probabilities are subsequently refined in real time during transmission using a sliding window approach, allowing the decoder to adapt to varying channel conditions and maintain high accuracy. In this study, we present simulation results that validate the effectiveness of the proposed model. The results demonstrate notable improvements in both error resilience and decoding accuracy, particularly as the density of MRAM increases.

Below is a summary of our contributions in this study:-We proposed a model that uses coding with a high code rate, which helps reduce the size of STT-MRAM while maintaining performance. This leads to minimized memory footprints, making the model suitable for embedded systems in the next generation of intelligent sensors.-With the proposed coding scheme based on the Hamming code, bit inversion, and a sliding window, the system can be designed simply for encoding and decoding. This helps reduce energy consumption and enables faster data transfer in sensor systems.-Finally, we provided simulation results to validate the feasibility and effectiveness of the proposed system.

The remainder of this paper is organized as follows. Section 2 provides an overview of STT-MRAM technology and introduces the associated cascaded channel model. Section 3 presents the proposed model for generating sparse codes based on the Hamming code theory, along with the corresponding decoding method using a syndrome-based algorithm. Section 4 offers an analysis of the simulation results and includes a comprehensive discussion of the findings. Finally, Section 5 concludes the paper with closing remarks and future directions.

## 2. Channel Model

### 2.1. Structure of STT-MRAM

The structure of an STT-MRAM cell is illustrated in Figure 1. An STT-MRAM device primarily consists of two components: a magnetic tunnel junction (MTJ) and an nMOS access transistor. The nMOS transistor regulates access to the memory cell, while the MTJ serves as the data storage element. The MTJ comprises two ferromagnetic layers separated by an ultrathin tunneling oxide barrier. One of these layers, known as the reference layer, has a fixed magnetization direction, whereas the free layer can change its magnetization in response to a spin-polarized current. The logical state stored in the MTJ is determined by the relative orientation of these two layers. When their magnetizations are antiparallel (AP), the MTJ exhibits a high-resistance state, representing a logical ‘1’. Conversely, when the magnetizations are parallel (P), the MTJ is in a low-resistance state, representing a logical ‘0’. STT-MRAM cells typically adopt a one-transistor-one-MTJ (1T1J) configuration, integrating the MTJ with an nMOS transistor for access control. The bit information is stored in the free layer and written by applying a spin-polarized current, which induces a magnetization switch. The fixed magnetization of the reference layer ensures a consistent comparison for resistance-based data sensing. The two resistance states—denoted as $R_{P}$ (parallel) and $R_{\mathrm{AP}}$ (antiparallel)—enable non-volatile read and write operations within the memory.

The writing process in an STT-MRAM cell is illustrated in Figure 2. To write a 0 and achieve the P state, a current is generated from the free layer to the reference layer (write-0 direction), as shown in Figure 2a. In this write-0 configuration, the voltage VDD is connected to the bit line (BL) and the word line (WL), while the ground (GND) is connected to the source line (SL). Conversely, to write a 1 and achieve the AP state, a current is driven from the reference layer to the free layer (write-1 direction), as shown in Figure 2b. In this case, the VDD is supplied to the SL and the WL, while the BL is connected to the GND. The reading process, shown in Figure 3, involves applying a small sense current through the MTJ. The resistance of the MTJ is measured using a dedicated memory-sensing circuit, which detects whether the cell is in the high-resistance (AP) or low-resistance (P) state, thereby determining the stored bit value. Notably, the direction of the current used during writing is also applicable in reading the stored data, although at much lower current levels to ensure non-destructive sensing.

In STT-MRAM, the reliability of memory cells is significantly influenced by process variations and thermal fluctuations, both of which contribute to write and read errors. Write errors occur when the applied switching current is insufficient to reliably complete the magnetization state transition of the magnetic tunnel junction (MTJ). Consequently, the memory cell may fail to switch correctly between logical states either from ‘1’ to ‘0’ (antiparallel to parallel) or from ‘0’ to ‘1’ (parallel to antiparallel). This issue is particularly pronounced during the 0 → 1 transition, which requires a higher switching current due to its inherently lower spin-transfer torque (STT) efficiency, compared to the 1 → 0 transition. As a result, the write error rate (WER) for the 0 → 1 transition ($P_{1}$) is significantly higher than that of the 1 → 0 transition ($P_{0}$), as demonstrated in previous studies [20,24,25]. For instance, the probability of failure during the transition from the parallel (P) to the antiparallel (AP) state is markedly greater than in the reverse direction. In addition to current-based limitations, thermal fluctuations further exacerbate the unreliability by introducing randomness into the MTJ’s magnetization process. These thermal effects increase the variability in switching behavior, thereby reducing overall write reliability and contributing to data integrity issues.

In STT-MRAM, read errors are generally categorized into two types: read decision errors and read disturbance errors. Read decision errors occur when the sensing circuit fails to accurately distinguish the resistance state of the magnetic tunnel junction (MTJ) during a read operation. These errors typically result from insufficient signal margins or variations in the resistance values, leading to incorrect bit detection. On the other hand, read disturbance errors arise when the applied read current is large enough to unintentionally alter the MTJ’s magnetic state.

### 2.2. Cascaded Channel

To effectively apply signal processing techniques in STT-MRAM systems, the use of a reliable channel model is essential. Such a model captures the behavior of the memory medium and the various impairments that degrade system performance. In this study, we adopted the cascaded channel model for STT-MRAM as proposed by Cai and Immink [17]. This model offers a systematic framework for analyzing the read/write processes and evaluating system performance under practical operating conditions. A brief overview of the cascaded channel model is presented below.

As mentioned earlier, write and read failure rates are key contributors to system performance degradation, significantly impacting system reliability. The parameters $P_{1}$ and $P_{0}$ denote the write error rates for 0 → 1 and 1 → 0 switching in the same order. The cascaded channel model combines the write error and read error models. The write error model is represented by a binary asymmetric channel (BAC), where the error probabilities for 0 → 1 and 1 → 0 are denoted as $P_{0}/2$ and $P_{1}/2$, respectively. The read error model is divided into two components: the read disturbance error and the read decision error. The read disturbance error is modeled by a Z-channel, with $P_{r}$ representing the crossover probability of 1 → 0 or 0 → 1 during the reading process in the write-0 or write-1 direction, respectively. The read decision error is modeled using a Gaussian mixture channel (GMC), where a Gaussian variable $R_{0}$, with mean $\mu_{0}$ and variance $\sigma_{0}$, represents low resistance and $R_{1}$, with mean $\mu_{1}$ and variance $\sigma_{1}$, represents high resistance. These three models (the write error model BAC, the Z-channel for read disturbance errors, and the GMC for read decision errors) are combined to form the cascaded channel model. Figure 4 illustrates the diagram of this cascaded channel model.

Furthermore, the write error model (BAC) and the read disturbance model can be combined to simplify the representation, as shown in Figure 5. The resulting crossover probabilities after the combination can be expressed as follows:For the write-0 direction:
(1)$$\;p_{0}=\frac{P_{0}}{2}\left(1-P_{r}\right);\;p_{1}=\frac{P_{1}}{2}+\left(1-\frac{P_{1}}{2}\right)P_{r};\;q_{0}=\left(1-\frac{P_{0}}{2}\right)+\frac{P_{0}}{2}P_{r};\;q_{1}=\left(1-\frac{P_{1}}{2}\right)\left(1-P_{r}\right).$$For the write-1 direction:
(2)$$\;p_{0}=\frac{P_{0}}{2}+\left(1-\frac{P_{0}}{2}\right)P_{r};\;p_{1}=\frac{P_{1}}{2}+\left(1-P_{r}\right);\;q_{0}=\left(1-\frac{P_{0}}{2}\right)\left(1-P_{r}\right);\;q_{1}=\left(1-\frac{P_{1}}{2}\right)+\frac{P_{1}}{2}P_{r}.$$

## 3. Proposed Model

The proposed model is depicted in Figure 6. User data ***u*** $\in \;\left\{0,1\right\}$ are extended by appending three check bits designed to enforce sparse code characteristics, resulting in signal ***s***. This signal ***s*** is subsequently multiplied by the generator matrix **G** to generate the codeword. The sparsity of a generated codeword is evaluated by computing the total number of logical ‘1’s it contains. If the codeword does not satisfy the predefined sparsity criterion, it is inverted to ensure compliance. The resulting sparse codeword ***c*** is then stored on the MRAM channel. Prior to transmission, the initial probabilities of bit-0 and bit-1 occurrences within the codeword are determined using the proposed method. These probabilities serve as a baseline for threshold detection, enabling more accurate and adaptive decoding in the presence of channel asymmetries and noise. The received signal ***r***, obtained from the resistance measurements of the magnetic tunnel junction (MTJ) in the MRAM device, is decoded using an advanced syndrome decoding algorithm. This algorithm enhances threshold detection by continuously adjusting the probabilities of bit-0 and bit-1 during transmission, employing a sliding window technique to adapt to channel variations. The decoded signal $\hat{c}$, mirroring the structure of the sparse codeword ***c***, is derived from ***r***. The three check bits embedded in $\hat{c}$ are then examined to ascertain whether an inversion occurred during encoding, yielding the recovered signal $\hat{s}$. This step ensures precise reconstruction of the original data. Finally, the user data $\hat{u}$ are retrieved from $\hat{s}$ by stripping away the check bits. To demonstrate the robustness of this approach across diverse channel conditions such as the offset condition and the writing error probability $P_{1}$, simulation results are presented and analyzed.

### 3.1. Encoder

The proposed encoding process comprises two primary steps. First, three check bits, initially set to zero, are appended to the user data vector ***u***, forming an augmented vector ***s*** = [***u*** 0 0 0]. Next, the corresponding codeword ***c*** is generated by multiplying ***s*** with the generator matrix **G**, derived from the Hamming code, i.e., ***c*** = ***s*G**. If the resulting codeword ***c*** does not satisfy the sparse code conditions, it is inverted to produce a valid sparse codeword. This ensures that the codeword meets the predefined sparsity requirement, which is critical for error resilience in asymmetric channels such as MRAM. The detailed encoding procedure is outlined as follows:Parity matrix construction: We construct a parity matrix **P**, consisting of binary tuples with a weight greater than 1, where the final tuple is *p_k_*_–1_ = (111…111). Each binary tuple has a length of *m*, and the matrix **P** has dimensions (2*^m^* − *m* − 1) ×
*m*, written as $P_{\left(2^{m}-m-1\right)\times m}$.Generator matrix formation: The generator matrix **G** is assembled by combining the identity matrix **I** with the parity matrix **P**. Mathematically, **G** is represented as:
(3)$$G_{\left(2^{m}-m-1\right)\times \left(2^{m}-1\right)}=\left[I_{\left(2^{m}-m-1\right)\times \left(2^{m}-m-1\right)}\;|\;P_{\left(2^{m}-m-1\right)\times m}\right].$$
Augmented data vector: the user data vector $u=\left[b_{0}\;b_{1}\;b_{2}\ldots b_{k-4}\right]$ is appended with three zero check bits to form $s=\left[b_{0}\;b_{1}\;b_{2}\ldots b_{k-4}\;0\;0\;0\right]$, where $k=2^{m}-m-1$.Codeword generation: the codeword ***c*** is computed by multiplying ***s*** with **G**, expressed as:(4)$$c_{1\times n}=s_{1\times k}G_{k\times n},$$
where $n=2^{m}-1$.Sparsity check: the sparse code characteristic of the codeword $c=\left[b_{0}\;b_{1}\;b_{2}\ldots b_{n-1}\right]$ is evaluated using the formula:(5)$$\omega \left(c\right)=\sum_{i=0}^{n-1}b_{i}.$$If *ω*(***c***) < *n*/2, ***c*** is output as the final codeword. Otherwise, the process continues.Sparse code adjustment: to ensure the codeword is sparse, the following formula is applied:(6)$$c\;=\bar{c}$$
where $\bar{c}$ represents the bitwise inversion of ***c***.

From these steps, it is evident that the resulting sparse code achieves a code rate of $\left(k-3\right)/n$. Codeword ***c*** is guaranteed to maintain sparse code characteristics.

### 3.2. Threshold Detection

For a cascaded channel, the threshold resistance $\left(R_{\mathrm{th}}\right)$ is used to distinguish between two cell resistance states. Based on the following rule for hard decisions:If $y_{k}\leq R_{\mathrm{th}}$, then $x_{k}^{d}=0$.If $y_{k}>R_{\mathrm{th}}$, then $x_{k}^{d}=1$.

The bit error rate (BER) of the hard detected channel bits $x_{k}^{d}$ can be computed as:(7)$$\begin{array}{l} \;P_{b}=\mathrm{Pr}\left(x_{k}=0\right)\mathrm{Pr}\left(x_{k}^{d}\neq x_{k}\left|x_{k}\right.=0\right)+\mathrm{Pr}\left(x_{k}=1\right)\mathrm{Pr}\left(x_{k}^{d}\neq x_{k}\left|x_{k}\right.=1\right)\\ =P_{0}\left[q_{0}\mathrm{Pr}\left(y_{k}>R_{th}\left|{\hat{x}}_{k}=0\right.\right)+p_{0}\mathrm{Pr}\left(y_{k}>R_{th}\left|{\hat{x}}_{k}=1\right.\right)\right]+\left(1-P_{0}\right)\left[q_{1}\mathrm{Pr}\left(y_{k}<R_{th}\left|{\hat{x}}_{k}=1\right.\right)+p_{1}\mathrm{Pr}\left(y_{k}<R_{th}\left|{\hat{x}}_{k}=0\right.\right)\right] \end{array}.$$

The BER is defined as the probability that a transmitted bit is decoded incorrectly. It accounts for the likelihood of transmitting a ‘0’ or ‘1’, as well as the conditional probability of each being erroneously decoded. Formally, the BER is computed as a weighted sum of these error probabilities, based on the respective bit occurrence probabilities in the transmitted codeword. As described in Section 2, the GMC within the cascaded channel model represents noise as a combination of Gaussian distributions, each with distinct means and standard deviations for the transmitted bits. For hard decisions, the threshold $R_{\mathrm{th}}$ dictates the decoding: a received signal $y_{k}$ exceeding $R_{\mathrm{th}}$ is decoded as 1, otherwise as 0. The error probabilities are then expressed using the Q-function, which quantifies the likelihood of the noisy signal crossing the threshold, transforming the BER into a form with Q-function terms that reflect the impact of Gaussian mixture noise on decision errors:(8)$$P_{b}=P_{0}\left[q_{0}Q\left(\frac{R_{\mathrm{th}}-\mu_{0}}{\sigma_{0}}\right)+p_{0}Q\left(\frac{R_{\mathrm{th}}-\mu_{1}}{\sigma_{1}}\right)\right]+\left(1-P_{0}\right)\left[q_{1}\left(1-Q\left(\frac{R_{\mathrm{th}}-\mu_{1}}{\sigma_{1}}\right)\right)+p_{1}\left(1-Q\left(\frac{R_{\mathrm{th}}-\mu_{0}}{\sigma_{0}}\right)\right)\right].$$

To optimize the probability $P_{b}$ with respect to the threshold $R_{\mathrm{th}}$, we set the derivative $\frac{dP_{b}}{dR_{th}}$ to zero. This step identifies the critical point where $P_{b}$ is either maximized or minimized, balancing the contributions of the two Gaussian distributions. The resulting equation ensures that the threshold $R_{\mathrm{th}}$ optimally separates the two distributions, weighted by their respective probabilities $P_{0}$ and $P_{1}$, thereby achieving the desired performance metric for the system.(9)$$R_{\mathrm{th}}=\frac{-b\pm \sqrt{b^{2}-4ac}}{2a},$$
where$b^{2}-4ac>0$,$a=\left({\sigma_{0}}^{2}-{\sigma_{1}}^{2}\right)$,$b=2\left({\sigma_{1}}^{2}\mu_{0}-{\sigma_{0}}^{2}\mu_{1}\right)$,$c=2{\sigma_{0}}^{2}{\sigma_{1}}^{2}\ln\left(\frac{\sigma_{1}}{\sigma_{0}}\times \frac{Pq_{1}-q_{1}+Pp_{0}}{-Pq_{0}+p_{1}-Pp_{1}}\right)$.

The formula for calculating the threshold $R_{\mathrm{th}}$ includes a constant representing the probability of 0 and 1 in the codeword before going through the channel. To improve accuracy, we proposed a new algorithm to compute this probability more effectively. The detailed steps are outlined as follows:Let us define the structure of the codeword in our model. A codeword ***c*** of length ***n*** is divided into three parts: user data of length k−3, sparsity check bits of length 3, and parity check bits ***p*** of length ***m***:(10)$$c_{1\;\times \;n}=[u_{1\;\times \;\left(k-\;3\right)}\;\left|\;0\;0\;0\;\right|\;p_{1\;-m}].$$Count the number of 0 s (*z*) and the number of 1 s (*t*) in the parity matrix $P_{\left(2^{m}-m\;\times \;1\right)\;\times \;m}$ excluding the last three rows.Determine the probability of 1 and 0 in the codeword as follows:(11)$$P_{1}=\frac{Number\;1s\;of\;all\;codewords}{Total\;bits\;of\;all\;codewords}=\frac{\frac{2^{k}}{2}k+m\times 2^{z}\times \sum_{i=1,3,5,\ldots }^{t}\left(\begin{array}{c} t\\ i \end{array}\right)}{2^{k}\times n}$$
(12)$$P_{0}=1-P_{1}$$ where $\left(\begin{array}{c} t\\ i \end{array}\right)$ is the combination operator of *t* elements and the number of *i* combinations.

After determining the probabilities of 1 and 0 in the codeword, these values are applied to the formula above to initialize the threshold resistance $R_{\mathrm{th}}$, which serves as a reference for decoding in the proposed model. To further enable adaptive threshold detection during transmission, a sliding window technique is employed to dynamically update the probability of 0 ($P_{0}$), allowing the system to adapt to variations in the MRAM channel, as depicted in Figure 7. The sliding window operates over the entire user data stream, which consists of a sequence of bits derived from the received signal measured via the magnetic tunnel junction (MTJ) resistance in the MRAM device. This window of a fixed size is segmented into three distinct regions: the “previous sub-window”, which contains bits from the prior position of the window before it advances; the “new window”, which represents the current set of bits under analysis; and the “new sub-window”, which includes the newly incorporated bits as the window slides forward. As transmission progresses, the window moves in a sliding fashion, continuously updating its contents by adding bits from the new sub-window and discarding bits from the previous sub-window, thereby maintaining a real-time view of the bit stream. This mechanism is integrated into an advanced syndrome decoding algorithm, which leverages the updated probability to adjust the detection threshold, ensuring accurate decoding of the sparse codeword under varying channel conditions. The probability $P_{0}$ is computed as the ratio of the net number of 0s in the window to the total number of bits considered, using the formula:(13)$$P_{0}=\frac{N_{\mathrm{pw}}+N_{\mathrm{ns}}-N_{\mathrm{ps}}}{W},$$
where$N_{\mathrm{pw}}$ is the number of 0s in the previous window;$N_{\mathrm{ns}}$ is the number of 0s in the new sub-window;$N_{\mathrm{ps}}$ is the number of 0s in the previous sub-window;W is the total number of bits in the window.

### 3.3. Decoder

After applying threshold detection to convert the scalar signal received from the cascaded channel into a binary signal ***r’***, this binary signal is multiplied with $H^{T}$ to identify the error bit position in the error table. The error table enables correction of a single bit error in ***r’***. Once the error is corrected, the resulting codeword $\hat{c}$ is checked for its sparse code characteristics. Specifically, if the last three bits of $\hat{c}$ are any of the following: [0 0 0], [0 0 1], [0 1 0], or [1 0 0], these last three bits are removed to retrieve the user data (recovered signal) $\hat{u}$. Otherwise, $\hat{c}$ is inverted, and then, the last three bits are removed to obtain $\hat{u}$. Finally, the signals ***u*** and $\hat{u}$ are utilized to evaluate the BER performance.

## 4. Results and Simulation

For the simulation, a total of 540,000,000 user data bits (***u***) are used. Following the sparse code characteristic check and necessary adjustments, the codeword (***c***) is generated and stored in the MRAM. During the data retrieval phase, the received signal (***y***) is obtained by measuring the resistance of the MTJ in the MRAM, where errors may occur. To mitigate these errors, the proposed threshold detection and syndrome-based decoder are applied to the received signal to reconstruct the estimated user data ($\hat{u}$). The BER performance is then evaluated by comparing the original data (***u***) with the recovered data ($\hat{u}$).

In the initial simulation, we aim to identify the optimal parameters for the sliding window by conducting a comprehensive survey. Theoretical insights suggest that a larger window size may overlook subtle data variations, whereas a smaller one might fail to capture sufficient contextual information. To balance this trade-off, we performed simulations across a range of window sizes to determine the configuration that yields the best BER performance. Specifically, we evaluated our method using window sizes corresponding to 5000, 7000, 10,000, 13,000, and 15,000 block codes, with each block consisting of 63 bits. Additionally, the $R_{\mathrm{th}}$ is updated every 1000 block codes.

Figure 8 illustrates the BER performance of our method using various window sizes: 5000, 7000, 10,000, 13,000, and 15,000 block codes. The results indicate that a window size of 10,000 block codes achieves the best BER performance (i.e., the optimal point) at a $\sigma_{0}/\mu_{0}$ ratio of 8%. Furthermore, in the range of $\sigma_{0}/\mu_{0}$ ratios from 9.5% to 16%, the performance gap among different window sizes is negligible. We also conducted simulations for $\sigma_{0}/\mu_{0}$ ratios between 5% and 7%; however, the results were not significantly different from those at 8% and thus are not included in the figure. Based on these findings, we selected 10,000 block codes as the sliding window size for our proposed model.

To demonstrate the impact of window sliding in our approach, we conducted simulations comparing the proposed method with and without the sliding window in Figure 9. We can observe that the sliding window improves the BER performance of the model as the $\sigma_{0}/\mu_{0}$ ratio decreases. This is because a smaller $\sigma_{0}/\mu_{0}$ ratio indicates reduced Gaussian noise, which allows the sliding window to estimate the *Rth* value more accurately and stably. As a result, the improvement becomes more significant when the $\sigma_{0}/\mu_{0}\;$ratio is low.

In the next simulation, the proposed model is implemented using a sparse code rate of 54/63. For comparison, we also included the results of the 7/9 sparse code rate presented in [8], the 51/63 BCH model using the decoding in [26], and the 56/63 syndrome and low-density parity check (LDPC) described in [23]. The device parameters for STT-MRAM are adopted from [27]. Specifically, the write error rates are set as $P_{1}$ = 2 × 10^−4^, $p_{1}$ = 1.02 × 10^−4^, and $p_{0}$ = 10^−6^. According to [13], the area density of an STT-MRAM cell has a direct impact on its write error rate. Therefore, we adjusted the write error rate parameters for each coding scheme based on the scaling model proposed in [13]. To achieve the greatest impact on size, we selected the grey line representing a ‘1’ written in 10 ns as the application result in our proposed model. We assumed that no coding is used at 450 nm, as the size is expected to decrease when coding is applied. Additionally, sizes smaller than 450 nm exhibit significant variation, which further emphasizes the impact of size on the error rate.

To calculate the error rate *P*_1_ in our paper, we took a example as below.

Using the 7/9 method, 450*7/9 = 350 nm. To maintain the same area density of user bits in this method, the size of each MTJ must be reduced. This allows more MTJs to be added within the same area compared to the no-coding method. Assuming 450 nm for the no-coding method and 350 nm for the 7/9 method, the authors of [13] indicate that this size reduction leads to a 25% increase in the error rate. We typically used *P*_1_ = 2 × 10^−4^. Therefore,
For the no-coding method: *P*_1_ = 2 × 10^−4^.For the 7/9 method: *P*_1_ = 2 × 10^−4^
× 1.25 = 2.5 × 10^−4^.

The updated write error rates applied in the simulation for each method are summarized in Table 1.

Figure 10 illustrates the BER performance of the proposed method in comparison with those of previously reported techniques, incorporating updated write error rates. The results indicate that the proposed method outperforms those reported in previous studies. At a BER of 10^−4^, the proposed method achieves a 2.8% improvement over the 7/9 sparse code. At a BER of 10^−6^, it shows a 0.5% gain over the 56/63 sparse code. However, when the $\sigma_{0}/\mu_{0}$ ratio increases to 16%, the BER performance of the proposed method falls slightly behind that of the 56/63 sparse code, although it still maintains a clear advantage over the 7/9 sparse code.

In the next simulations, we adopted the methodology described in [28] to evaluate performance under varying offset conditions. Specifically, the environment was configured with a $\sigma_{0}/\mu_{0}$ ratio of 9.5%, and the mean offset $\mu_{off}$ was varied from −0.25 to −0.05. The corresponding results are presented in Figure 11 and Figure 12, which illustrate the BER performance for $\sigma_{off}/\mu_{1}$ ratios of 5% and 7%, respectively. As shown in Figure 11 and Figure 12, the proposed model demonstrates superior performance compared to previous studies.

Finally, we set $\sigma_{0}/\mu_{0}=\sigma_{1}/\mu_{1}=9.5\%$ and adjusted the writing error probability $P_{1}$ (error rate for the 0 → 1 transition) during the writing process [29]. The performance of the proposed model is presented in Figure 13. As illustrated, each model exhibits a distinct threshold error probability, beyond which the BER begins to degrade from its previously stable value, indicating a loss in decoding reliability as write error rates increase. In addition, the proposed code can still improve the performance. Moreover, when $P_{1}$ is less than 10^−5^, which is very small, the error in the writing process does not affect the performance of the codes, but the read-decision error occurs at the GMC with the resistance spread $\sigma_{0}/\mu_{0}=\sigma_{1}/\mu_{1}=9.5\%$.

To compare the complexity of our proposed decoder with that of previous studies, we quantified and grouped both addition and comparison operations as equivalent addition operations for the purpose of complexity analysis. Table 2 presents the results of the complexity comparison.

In Table 2, we reference the complexities of the 7/9 mapping, the 56/63 sum-product, and the 54/63 syndrome (without the sliding window), as presented in [23]. Our proposed model is based on the 54/63 syndrome method from [23] and, therefore, achieves a similar level of complexity. The primary distinction between our model and the 54/63 syndrome in [23] lies in the use of sliding window operators. Compared to the 7/9 mapping, our model represents a trade-off between performance and complexity. In contrast to the 56/63 sum-product method, our model reduces complexity, as the sum-product algorithm relies on iterative computations, which significantly increase the number of required operations.

To calculate the complexity of our proposed model, we began with the decoding model of the 54/63 syndrome and added the operators introduced by the sliding window mechanism. The number of sliding window operators was estimated based on Equation (13), which was used to count the total number of addition operations across 10,000 data blocks. The value of *Rth* obtained from these 10,000 blocks was then used to detect the next 1000 data blocks, using a step size of 1000 blocks (i.e., the previous sub-window size was 1000). Therefore, the total number of operators for the 10,000 blocks was divided by 1000 to determine the average number of operations per data block. Since each data block consisted of 63 bits, the number of operations per block was further divided by 63 to compute the number of operations per detected bit. This final value was then added to the number of operators used in the 54/63 syndrome decoding from [23].

## 5. Conclusions

In this paper, we presented enhancements to both the encoding and decoding processes to overcome the limitations of existing sparse coding schemes. By introducing additional check bits during encoding to enforce sparse code characteristics, we ensured that the resulting codewords exhibit the desired sparsity. Furthermore, we improved the syndrome decoding mechanism by incorporating a dynamic threshold detection method that adaptively updates the probabilities of bit-0 and bit-1 throughout the transmission process. This adaptive approach significantly improves decoding accuracy and robustness under varying channel conditions. Simulation results demonstrate that the proposed scheme achieves a higher code rate and superior BER performance compared to existing methods, thereby validating its effectiveness in addressing reliability challenges in STT-MRAM systems. Future research may investigate further optimizations and the applicability of this approach to other emerging memory technologies. In addition, the proposed method relies solely on simple techniques, such as the Hamming code for encoding and decoding, bit inversion for sparse modulation, and a sliding window for updating bit probabilities. Given the performance and simplicity of the proposed model, it has the potential to be implemented efficiently and with low overhead in future systems.

## Figures and Tables

**Figure 1 sensors-25-04050-f001:**
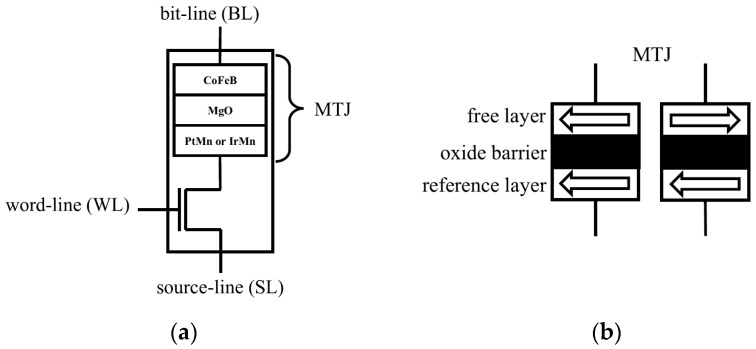
(**a**) STT-MRAM cell; (**b**) parallel state (PS) and antiparallel state (AS).

**Figure 2 sensors-25-04050-f002:**
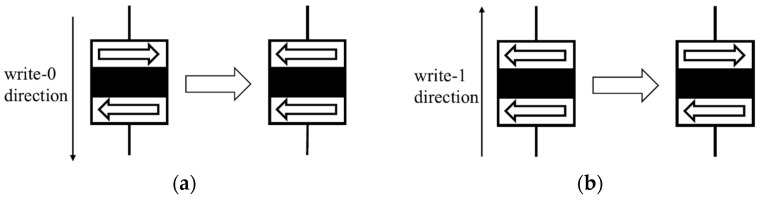
Writing process. (**a**) 1 to 0 switching. (**b**) 0 to 1 switching.

**Figure 3 sensors-25-04050-f003:**
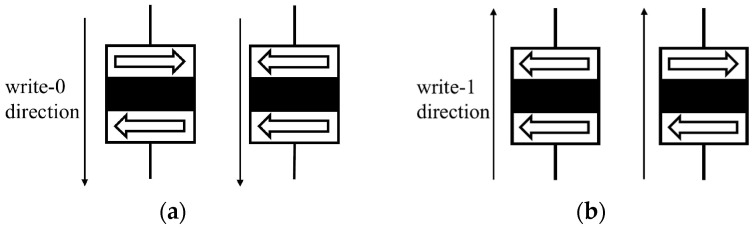
Reading process. (**a**) Reading with a write-0 current. (**b**) Reading with a write-1 current.

**Figure 4 sensors-25-04050-f004:**
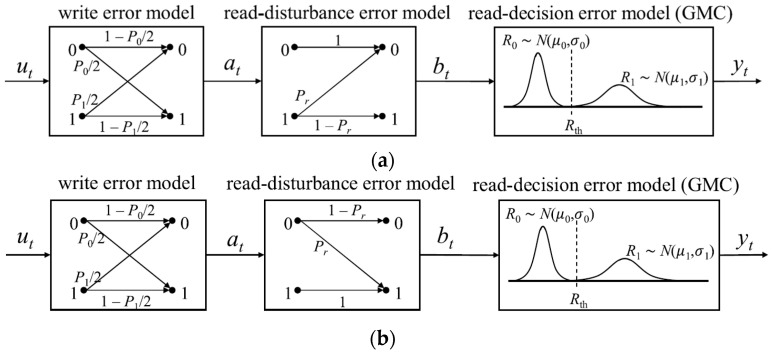
Cascaded channel model. (**a**) Reading with a write-0 direction. (**b**) Reading with a write-1 direction.

**Figure 5 sensors-25-04050-f005:**
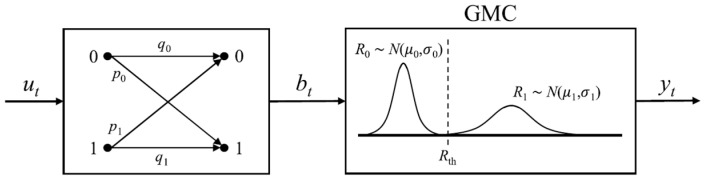
Cascaded channel combination model.

**Figure 6 sensors-25-04050-f006:**
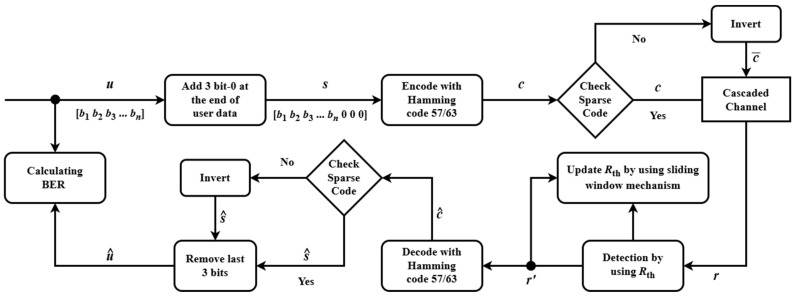
Diagram of the proposed model.

**Figure 7 sensors-25-04050-f007:**
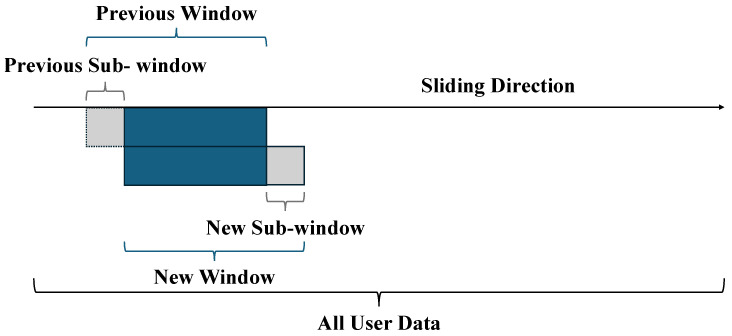
Illustration of the sliding window technique used to update the probability of 0 ($P_{0}$) during transmission in the proposed model.

**Figure 8 sensors-25-04050-f008:**
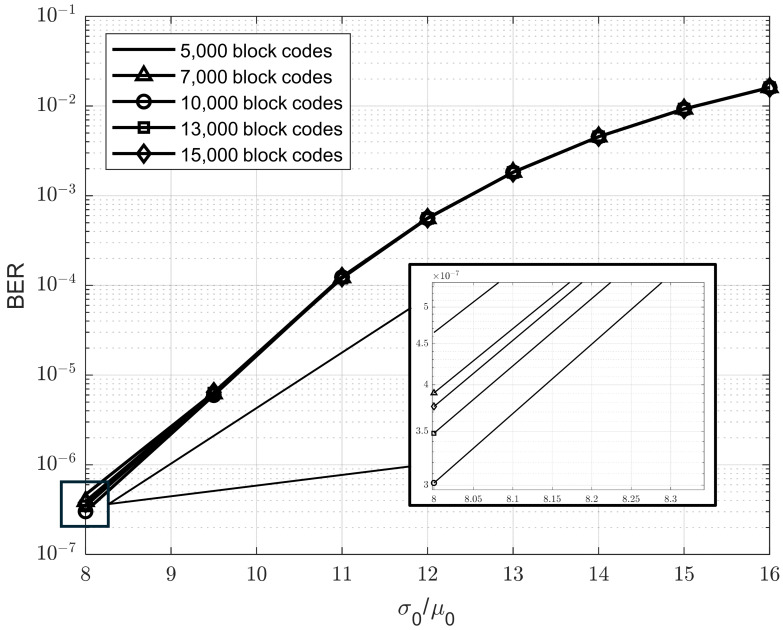
BER performance comparison across different window sizes.

**Figure 9 sensors-25-04050-f009:**
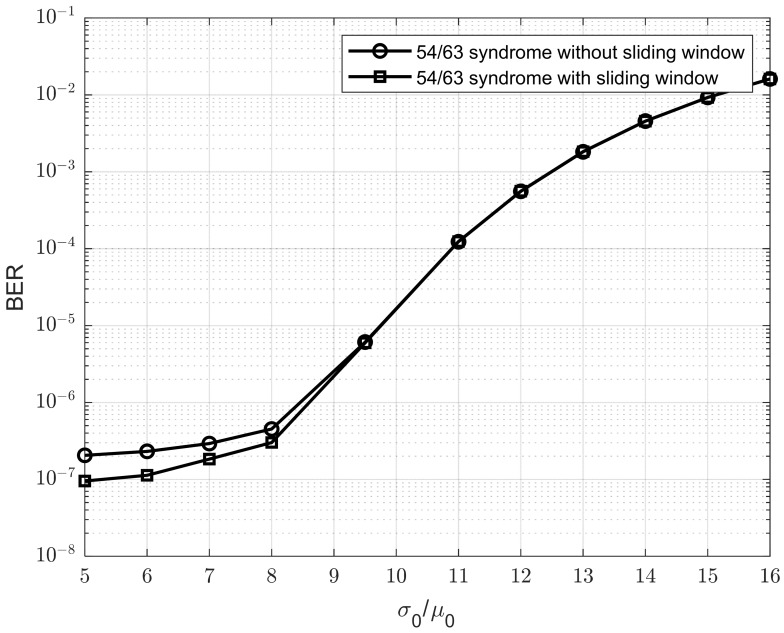
BER performance comparison between the proposed model with and without the sliding window.

**Figure 10 sensors-25-04050-f010:**
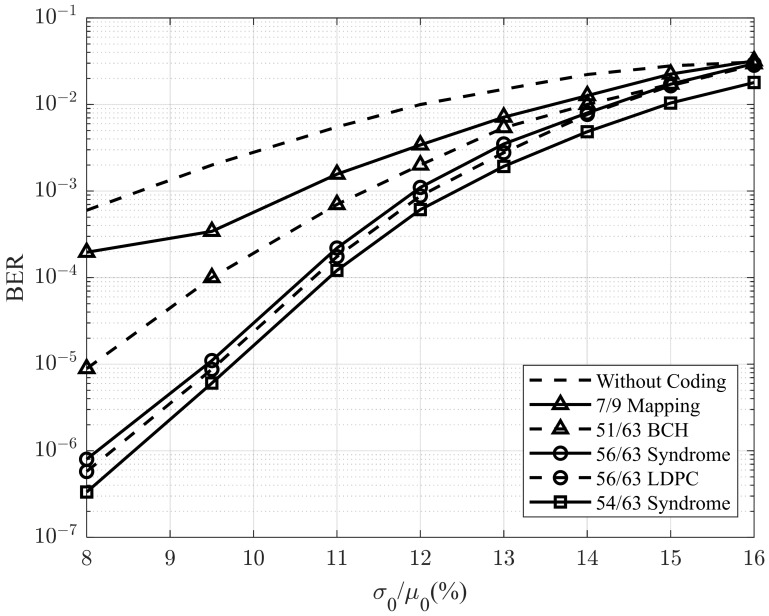
BER performance of the proposed method and the previous methods.

**Figure 11 sensors-25-04050-f011:**
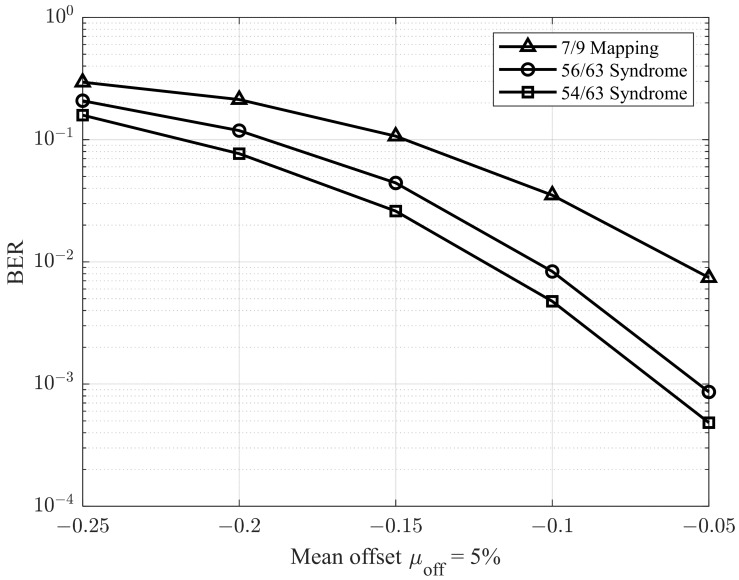
BER performance with different mean offsets $\mu_{off}$ and $\sigma_{off}/\mu_{1}=5\%$.

**Figure 12 sensors-25-04050-f012:**
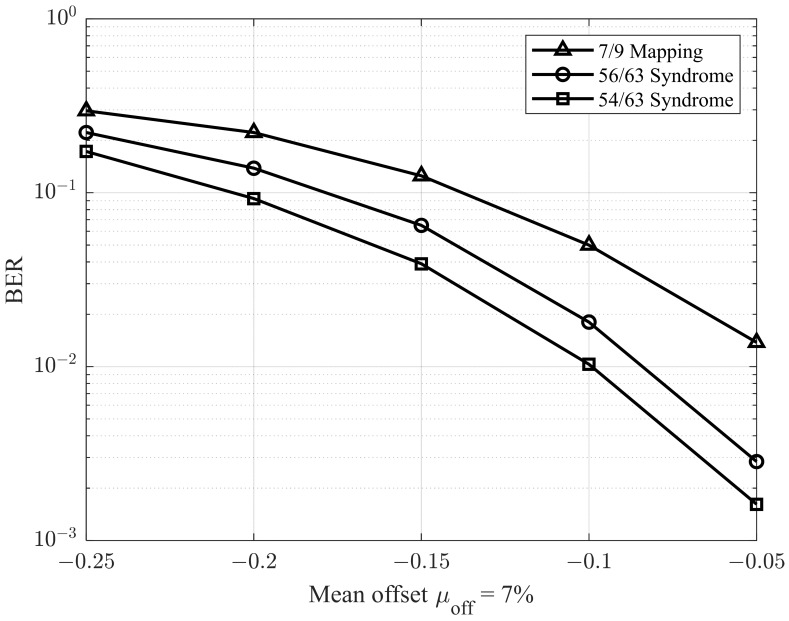
BER performance with different mean offsets $\mu_{off}$ and $\sigma_{off}/\mu_{1}=7\%$.

**Figure 13 sensors-25-04050-f013:**
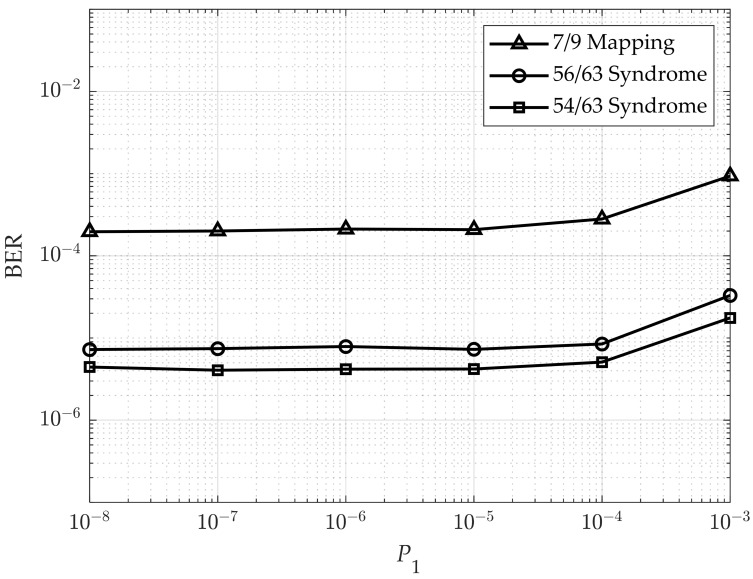
BER performance according to write error rate $P_{1}$.

**Table 1 sensors-25-04050-t001:** The write error rates of ‘1’ and the transistor channel widths based on the code rate after the update.

Code Rate	Transistor Channel Width (nm)	$\mathrm{Write}\;\mathrm{Error}\;\mathrm{Rate}\;\text{‘}1\text{’}\;(P_{1})$ in 10 ns
No coding	450	${2\;\times \;10}^{-4}$
7/9 method	350	${2.5\;\times \;10}^{-4}$
56/63 method	400	${2.1\;\times \;10}^{-4}$
54/63 method	385.7	${2.2\;\times \;10}^{-4}$

**Table 2 sensors-25-04050-t002:** Complexity corresponding to code rates and decoders.

Code Rate	Add/Sub	Mul/Div
7/9 mapping	256	128
56/63 sum-product	3830	5030
54/63 syndrome without the sliding window	441	378
54/63 syndrome with the sliding window	441.417	378

## Data Availability

The data presented in this study are available on request from the corresponding author.

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
