# Peer review of "Improving MRAM Performance with Sparse Modulation and Hamming Error Correction"

_sensors, 2025, doi:10.3390/s25134050_

Round 1
Reviewer 1 Report
Comments and Suggestions for Authors The authors successfully report the Improving MRAM Performance with Sparse Modulation and Hamming Error Correction in this study. The authors proposed a novel sparse coding scheme characterized by a minimum Hamming distance of three. During the encoding process, three check bits are appended to the user data and processed using a generator matrix. If the resulting codeword does not satisfy the sparsity constraint, it is inverted to meet the coding requirements. In addition, the authors introduced a dynamic threshold detection technique that updates bit probability estimates in real time during data transmission. Simulation results demonstrate that as MRAM density increases, this technique significantly improves both error resilience and decoding accuracy.The reviewer believes this article presents an innovative and relevant topic within its field. It addresses existing research gaps and contributes new findings to the related area of study.
Compared to other published papers, the reviewer finds that this article contributes a novel sparse coding scheme to the subject area. The proposed method is characterized by a minimum Hamming distance of three. During the encoding process, three check bits are appended to the user data and processed using a generator matrix. If the resulting codeword does not satisfy the sparsity constraint, it is inverted to meet the coding requirements. In addition, the reviewer has not identified any specific areas for further improvement in this article.
Finally, the references, tables and figures in this article were appropriate. In this article, the reviewer believes that the conclusions presented by the authors are highly valuable for future advancements in MRAM density, as well as for improving error resilience and decoding accuracy in related technologies. Therefore, the reviewer recommends that the article be accepted and published in the journal.
Reviewer 2 Report
Comments and Suggestions for Authors
Summary
The authors propose a sparse error correction scheme for MRAM using Hamming codes. They assume asymmetric error rate between writing of 0 and 1 values and evaluate the effectiveness through simulation of fixed error distributions.
Major Issues
- The proposed window function to update the error probability/read threshold is not covered by the evaluation. Neither is it discussed nor are possible parameter ranges established or tried. For Figure 11 the authors mention "adapting the error rate during the writing process" but don't discuss it further. Without discussion or evaluation, the sliding window feature is meaningless. No example/reference set of parameters for the sliding window is provided. Since the authors establish the sliding window approach as their key improvement upon [24], what value does the current work present without it?
Without a thorough evaluation of the window function, the paper is inacceptable! - The evaluation would benefit from comparison against more candidate error correction schemes as it is lacking in depth. "across diverse channel conditions" as written by the authors would require more diverse analysis. For example, a time variance of error distribution would highlight up- and downsides of the method. Likewise, a discussion of the efficiency and error correction vs implementation overhead would benefit the work greatly.
Minor Issues
- Minor grammatical issues in text/captions (e.g. Table 1 "and transistor channel width based on code after updated").
- References 2 and 7 are identical.
- "Reverse" and "Invert" are used interchangeably at times (e.g. in Figure 6). Yet, particularly for bit patterns, they have different meaning.
- In parts very repetitive. Especially for the explanation of the asymmetric error rates.
- Page 1: Line 20: The proposed mechanisms do not exploit the error characteristics, they simply want to counteract them.
Reviewer 3 Report
Comments and Suggestions for Authors
This work developed a sparse Hamming code scheme with dynamic threshold detection for STT-MRAM by a codeword-inversion strategy. The scheme exhibited continuously tunable sparsity with evident suppression of asymmetric bit errors. The authors believed that the synergistic effect of sparse modulation and adaptive probability estimation can suppress dark current noise, which is beneficial for high-density MRAM. They achieve both improved BER performance (2.8% gain at 10⁻⁴ BER) and robustness to write errors/offsets. The manuscript is well organized. This paper can be accepted after addressing the following comments.
- Lack of quantitative comparison with traditional error-correcting codes (such as BCH, LDPC, etc.).
- The criteria for selecting the sliding window size W have not been specified. It is necessary to demonstrate the impact of different W values on the BER.
- Other details issues
- Reference 7 is repetitive of Reference 2; Reference 17 has an unorganized format.
- In Figure 8, the data label obscures the data.
Round 2
Reviewer 2 Report
Comments and Suggestions for Authors
- Summary of contributions in the introduction: The claims of the paper are in some cases spread over multiple statements that would be better joined into one statement. Examples: Authors claim both "model that reduces the size of STT-MRAM" and "proposed codeword improves the code rate". This could be rephrased as one statement. They claim the "lightweight techniques used" can either allow an "efficient on-chip implementation" (first point) or "strong potential to reduce energy consumption" (third point). These claims also could be joined into one statement.
- Response to comment 1: The comparison of performance with/without sliding window in the revised paper shows a minor benefit of the sliding window method, in the range of bit error rates below 10^(-6) and only for 𝜎_0/𝜇_0 between 8 and 9.5. Compared to [23], the additional influence of the window, especially considering the additional complexity of the method (read hardware effort), is very small. The analysis of different window sizes provided by the authors shows that the window size does have an influence on the performance of the solution, but again in a narrow range.
To make this contribution meaningful, the authors should consider to evaluate a situation where the adaptivity of the window function is showing some true meaningful benefit (e.g. a jump in underlying condition, shown over time or a gradual deterioration, as hinted at in Comment 2). In the static case, it makes little difference and does not justify a new publication.
Response 2: When comparing Figure 10 of the current version of the paper with a similar graph in [23], there seems to be some inconsistency. In [23] the different curves are much closer to each other. The reason for this seems to originate from the recalculated error rates in Table 1. The recalculation is not justified enough in the paper. It looks as if it was only done to improve the effect of the method. The applicability of the scaling ruies from [13] is not discussed and needs to be shown.
